# The Memory Abilities of the Elderly Horse

**DOI:** 10.3390/ani14213073

**Published:** 2024-10-25

**Authors:** Syria Cellai, Angelo Gazzano, Lucia Casini, Valentina Gazzano, Francesca Cecchi, Fabio Macchioni, Alessandro Cozzi, Lucie Pageat, Sana Arroub, Sara Fratini, Martina Felici, Maria Claudia Curadi, Paolo Baragli

**Affiliations:** 1Department of Veterinary Sciences, University of Pisa, 56126 Pisa, Italy; syria.cellai@gmail.com (S.C.); angelo.gazzano@unipi.it (A.G.); lucia.casini@unipi.it (L.C.); francesca.cecchi@unipi.it (F.C.); fabio.macchioni@unipi.it (F.M.); maria.claudia.curadi@unipi.it (M.C.C.); paolo.baragli@unipi.it (P.B.); 2IRSEA Research Institute in Semiochemistry and Applied Ethology, 84400 Apt, France; a.cozzi@irsea-institute.com (A.C.); l.pageat@irsea-institute.com (L.P.); s.arroub@irsea-institute.com (S.A.); 3Department of Biology, University of Florence, 50121 Firenze, Italy; sara.fratini@unifi.it; 4Department of Agricultural and Food Sciences, University of Bologna, 40126 Bologna, Italy; martina.felici6@unibo.it; 5Bioengineering and Robotic Research Centre “E. Piaggio”, University of Pisa, 56126 Pisa, Italy

**Keywords:** horse, ageing, behavioral test, memory

## Abstract

The lifespan of domestic animals has progressively increased, leading to the emergence of organic and behavioral pathologies during senescence that were not previously observed in pets. In humans and other animal species, issues related to both short-term and long-term memory frequently arise, even during the normal aging process. However, age-related declines in short-term and long-term memory have not yet been studied thoroughly in horses. In the present research, the use of a behavioral test based on touching a target (Target Touch Test) demonstrated that even elderly horses can learn new associations between stimuli and maintain their memory even after 10 days. However, they show slower recovery times for recalling memorized information compared to animals under 16 years of age.

## 1. Introduction

The lifespan of domestic animals has progressively increased in recent years [1], following improvements in their management and the availability pf more effective therapies for their pathologies. Senescence requires addressing organic and behavioral pathologies that previously did not manifest in pets. Aging is a multifactorial process that leads to a decline in the functions of most organs and tissues [2]. A new discipline, “geroscience”, refers to research aimed at understanding the mechanisms of biological aging [3]. Although the main changes that accompany the aging process have been identified, the course of aging in individuals can be variable and unpredictable, determined by numerous interactions between environmental factors and living organisms [4,5,6]. In addition to these purely organic aspects, aging influences cognitive, emotional, and social aspects both in humans and animals [7]. 

Cognitive decline can manifest in various degrees of severity, ranging from the normal age-related decrease in cognitive abilities to pathological conditions such as dementia. Among domestic animal species, dogs have been the subjects of research due to the similarities between their aging process and that of humans, making them a promising experimental model [8]. There are many studies on the cognitive decline of dogs because of its parallels with human Alzheimer’s disease [9,10,11,12,13], which is also linked to inbreeding [14,15]. In selective breeding, inbreeding is commonly used to reinforce desirable traits in pure-breed animals, but it can have negative effects on animal welfare [16] and may lead to decreased selection response for economically important traits in all species, including dogs [17] and horses [18].

In recent years, increasing attention has been paid to horses, which are progressively being recognized as pets in addition to their roles in sports. In 2007, a survey of over 47,000 households across the USA showed that 56.5% of the people who filled out the questionnaire considered the horse to be a pet as they did dogs and cats, and 38.4% of horse owners considered their animals to be family members [19]. Consequently, it is becoming more common for these animals to reach advanced ages and exhibit the signs of physical and cognitive decline typical of aging. There is a quite a general consensus that a horse’s old age begins around 15 years, when reproductive performance starts to decline [20,21,22,23,24,25,26,27]. However, horses often live into their 30s, and some have reached and exceeded 40 years of age [28,29,30].

Increasing age is associated with a higher incidence of specific pathologies. Research conducted on 467 horses over the age of 20 reported frequent gastrointestinal, musculoskeletal, and respiratory tract problems [31]. In both humans and other animal species, problems related to short-term and long-term memory are commonly observed during the physiological processes of senescence. Jonker and colleagues, in their review of clinical and population-based studies, estimated the prevalence of memory complaints to be between 25 and 50% in people over the age of 65, with a higher incidence in women [32]. In canine species, Head and colleagues [33] found that spatial learning and memory were sensitive to age. Their study revealed that a significantly higher proportion of aged dogs were unable to learn a spatial delay non-matching-to-sample task.

Different classifications of memory are reported in the literature. According to content, memory is classified as explicit or implicit [34]. Considering the nature of memory, it can be categorized into archival versus transient, moment-to-moment memory [35,36]. Finally, based on duration, memory can be classified into short-term and long-term memory [37,38]. Short-term memory develops within seconds or minutes and lasts for several hours, while long-term memory undergoes slow consolidation [34,39]. Memories that last at least 24 h are considered long-term memories [40]. There are many factors that influence memory: the contribution of emotions [41] and REM sleep [42] are two elements that have been studied in animals. The study of memory is therefore very difficult and further complicated by the reliability of the tests used to measure it, which can provide results that are influenced by the individual’s motivation [43]. However, the evaluation of the latency with which an individual expresses a learned behavior is considered a reliable measure of memory, and this is a technique widely used in behavioral tests based on passive avoidance [44,45] or to evaluate memory with the Morris water maze [46].

While memory and its decline have been widely studied in canine species [47,48], a limited amount of research has been conducted on short- and long-term memory in horses, [49,50,51,52], and data on memory decline with advancing age in this species are lacking. 

Studying memory decline in elderly horses and identifying tools for its evaluation can improve the management of these animals and safeguard their welfare even in old age. Therefore, the aim of the present research was to develop a simple and reliable behavioral test to assess how short- and long-term memory are affected by aging in the equine species. For this research, short-term memory was defined as memory that lasts at least 30 s [50], while long-term memory was defined as a memory that has been retained after 10 days [53,54].

## 2. Materials and Methods

For the experiment, 44 clinically healthy horses (27 females and 17 geldings) in good nutritional condition were enrolled from three riding schools. None of the animals enrolled in the test showed abnormal behaviors or behavioral stereotypies. They were divided into two age groups: adults [5–15 years; n° = 21; mares = n° 12 (57.14%), geldings = n° 9 (42.86%)] and seniors [>15 years; n° 23; mares = n° 15 (65.22%), geldings = n° 8 (34.78%)]. Table 1 reports the age, sex, and breed data for each animal.

The general routines of the subjects remained unchanged, and they received ad libitum access to forage and water with an integration of concentrated feeds (at 7 a.m. and 6 p.m.) appropriate for their needs. The horses had regular access to outdoor paddocks (from 8 a.m. to 6 p.m.) and a workload tailored to their age and physical ability.

The tests were conducted initially (T1) and repeated after ten days (T10). All experiments were carried out in May 2024 under the same environmental conditions (time of the day from 5 p.m. to 6 p.m., stable management, and foraging) to minimize confounding factors. The horses were subjected to a behavioral test, which involved conditioning them to a clicker, used as a bridging reinforcer to reward the animals when they touched a target with their muzzle. The latency in touching the target immediately after the conditioning phase and 10 days after allowed us to evaluate the animal’s short- and long-term memory.

### Target Touch Test

First Phase: At the beginning of the experiment, each horse was placed in the box where they were usually stabled (5 m × 4 m) and given 15 min to settle in. Next, the horse was conditioned to associate the sound of the clicker with positive reinforcement. None of the horses had ever been subjected to this type of conditioning before. As a reinforcement, a piece of carrot was used, since this was a food that the owners occasionally gave to the animals as a reward that was extremely appreciated. The operator positioned himself in front of, to the right of, and to the left side of the horse, triggering the clicker and delivering the reward. After these three “clicker-reinforcement” associations, the operator moved to a position behind the horse on the left side, made the clicker sound, and observed whether the horse turned its head to receive the reinforcement. If the horse did not respond correctly, the procedure was repeated.

Second Phase: A target, consisting of a 68 cm long stick with a yellow tennis ball at the end, was presented to the horse by the operator. The target was held approximately 50 cm from the horse’s nose and presented five times consecutively, with the position varying inside the box, always starting from the left side of the horse. The clicker was activated, and the reward was given only if the horse touched the target with its muzzle.

Third Phase: Thirty seconds after the end of the second phase, the target was placed against one of the walls of the box (away from the food container and the door) approximately 50 cm from the horse’s nose. The experimenter then moved away from the target to ensure that the horse had no cues to guide it toward the target. The horse had to be able to see where the target was placed and had up to 180 s to touch the target to pass the test. If the horse touched the target, the clicker was activated, and the reward was given. If the horse did not perform the test within the established time, the maximum time available was recorded.

Regardless of the result, the second and third phases were immediately repeated two more times in the same way. The approach times for each of the three trials during the third phase were recorded.

Ten days after the initial test, the horse was tested again under the same environmental conditions. The animal was placed in the box and the experimenter placed the target on the wall, 50 cm from the horse’s nose. The reward was granted only if the animal touched the target within 180 s. This test was repeated three times, with a one-minute interval between trials, and each test completion time was recorded.

Statistical analysis was performed using Jamovi (Version 2.5) retrieved from https://www.jamovi.org (The Jamovi project 2024), accessed on 25 June 2024. The Shapiro–Wilk test was used to assess the normal distribution of the data. The Friedman test (a non-parametric test for repeated measures) and the Durbin–Conover test (for pairwise comparisons) were employed for statistical analysis.

## 3. Results

All selected horses successfully completed the first phase of the test, quickly associating the sound of the clicker with a reward after just three pairings. The dataset concerning to the times taken in the individual tests are provided in the Appendix A.

Figure 1 shows the times taken by adult and senior horses to touch the target in the three consecutive tests conducted at T1. A statistical analysis, using the non-parametric Friedman test, revealed no significant differences in the times taken by adult and senior horses across these three tests.

With regard the time taken by the adult and senior horses to touch the target at T10, the results of the three consecutive tests are shown in Figure 2.

Statistical analysis, using the non-parametric Friedman test, indicated a significant difference (χ² = 42.1; *p* < 0.001) in the times taken by adult and senior horses across the three tests at T10. This statistically significant difference was not attributable to the different times taken by adult or senior subjects to complete the three repetitions (a, b, c) of the test. Using the Durbin–Conover test, it was evident that there were statistically significant differences between adult and senior horses. Aged horses appeared to take significantly longer to complete the three repetitions of the test, as can be seen from the data reported in Table 2.

Notably, two horses in the senior group failed to complete the test in any of their three attempts.

Regarding the influence of sex on performance, Figure 3 and Figure 4 show the times taken by the adult and senior horses of both sexes to complete the Target Touch Test at T1 and T10.

Statistical analysis revealed significant differences between adult and senior horses (χ^2^ = 56.3; *p* = <0.001), with longer times taken by senior horses (male and female) at T10, but the Durbin–Conover test showed no statistically significant difference between the times taken by male and female adult and senior horses to complete the three single tests (a, b, c) at T1 and T10.

## 4. Discussion

Learning is a crucial process for all living beings, enhancing their ability to adapt to the environment and thus increasing individual fitness [55]. In domestic animals, which often have close interactions with humans and are involved in various activities such as hunting, horse riding, herding, and sporting activities, learning specific behaviors and rules is essential for developing a satisfactory human–animal relationship. This is especially true for the horse, whose learning abilities have been the subject of research several times [56,57,58,59], being trained by humans for over five millennia [60,61].

All learning is fundamentally based on the individual’s ability to store information in memory and recall it when needed.

The recent literature includes some research on the horse’s memory capacity [49,50], but studies on its decline with age are lacking. The results of this research provide a first step in evaluating cognitive aging in horses, particularly in terms of learning speed and memory retention and recall.

In this research, two training methods were used: clicker training and targeting training. Clicker training involves a hand-held device that makes a clicking sound when pressed during a desired behavior, typically followed by a reward [62,63]. According to Skinner’s reinforcement hypothesis [64], the clicker serves as a secondary reinforcer, ac-quiring the reinforcing properties of the primary reinforcer (usually a food reward) with which it is paired. Thus, in the first phase of the experiment, the horses were conditioned to associate the click with the reward.

Targeting involves teaching the animal to touch an object with a part of its body (usually nose, front leg, or hind leg) to receive a reward.

Both methods are part of the “gentle” approach to animal training, avoiding the use of punishments or negative reinforcements to achieve desired behaviors [65].

One of the main challenges in assessing memory is identifying an appropriate test that is not influenced by fluctuations in the individual’s motivation [43], which can significantly affect performance. In this study, a method based on evaluating the latency in the expression of the desired behavior was used to assess the individual’s memory capacity. This method, which has been widely used in the past with other species [44,45,46], showed in this study that it can be successfully applied to equine species as well. To eliminate the effects of changes in motivation while maintaining the reliability of the results, the number of test repetitions was limited to three.

The target test, which was created for this experiment, proved to be easily applicable to horses, even in non-laboratory experimental conditions. The speed of execution and the ease of preparation of the experimental setup are the most innovative aspects of this test that make it applicable to a large-sized species such as the horse, with which the researcher often must work, adapting to the environmental situation in which the animal is kept. An expert operator can carry out the test without influencing the animal, although the presence of the person is a criticality to be considered. The use of an automated device could eliminate this problem, but it would require a longer training of the animal.

In our test, all the horses quickly associated the click with the reward in phase 1, demonstrating that even senior horses are capable of associative learning. In phase 3 of the test at T1, senior horses managed to touch the target in times comparable to those of adult horses, with no significant differences observed across the three tests.

Additionally, no significant differences were found in the times taken by males and females, both adults and seniors, in the three tests at T1. These results suggest that the methodology used is effective in producing robust learning in horses, even at an advanced age, and is suitable for testing short-term memory.

Previous research on horses’ short-term memory has yielded conflicting results. A study by McLean suggested that horses could not remember the correct goal choice 10 s after the presentation of an associated cue in a flexible choice context [49], while a more recent study by Hanggi [51] demonstrated that horses could recall the location of a food objective in a two-choice response test under delays ranging from 5 to 30 s. Based on our data, we conclude that horses retain memory for at least 30 s and that senior horses maintain good learning and short-term memory abilities. The times taken to touch the target at T1 were not in fact statistically different between the adult and senior horses. 

However, the situation was different when long-term memory was considered. Previous studies have shown that horses can recall a learned response in a maze after one week [53] and can remember correct choices in multiple two-choice discrimination tasks for several months [54]. Our data show that two senior horses failed to complete the test in any of their three attempts at T10, and each attempt by the senior horses was much slower than the adults. This may suggest a greater difficulty in retrieving memorized information, a difficulty that did not decrease between the first and third attempts.

Regarding the influence of sex, our data suggest no significant difference between males and females in the completion times of the three tests, although great variability in the results of senior females was evident, especially in the test carried out at T10.

Further studies will be necessary to evaluate with greater certainty the differences in cognitive decline related to sex and whether those differences found in the human species, where it has been recently demonstrated that women experience faster cognitive decline than men, also occur in horses [66].

The small number of animals, belonging to many different breeds, prevented more detailed analyses and constitutes a limitation of the present research. It is also evident that the aging process presents strong individual differences that the test applied in this research may not have identified. Other factors also deserve a more in-depth analysis, such as the quantity and quality of sleep of these animals, considering the influence that the REM phase can have on the formation of memory [42]. Moreover, further studies will be needed to determine if cognitive impairment in horses is influenced by the types of activities they perform and their management.

## 5. Conclusions

The results of this research provide the first available data on the short- and long-term learning and memory abilities of elderly horses. A comparison with adult horses indicates that even older horses are capable of associative learning and can retain this learning for at least 10 days. The behavioral test used (Target Touch Test) proved to be straightforward to administer, effective in achieving a good level of learning in both adult and elderly horses, and suitable for assessing short- and long-term memory. Initial findings showed no significant differences in the memory abilities between mares and geldings.

## Figures and Tables

**Figure 1 animals-14-03073-f001:**
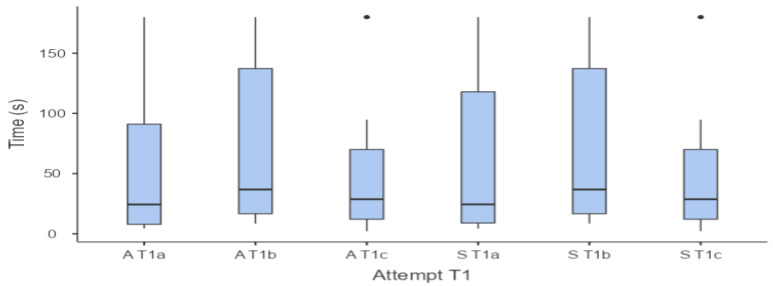
Box-and-whiskers plot reporting the times taken by adult (A; n = 21) and senior (S; n = 23) horses to touch the target at T1 in the three consecutive attempts (a, b, c).

**Figure 2 animals-14-03073-f002:**
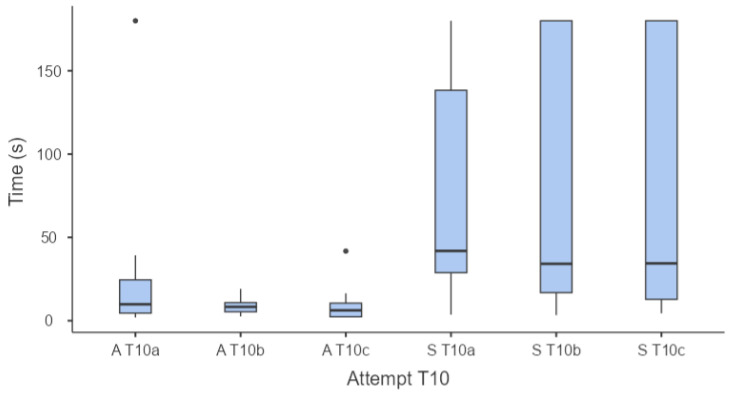
Box-and-whiskers plot reporting the times taken by adult (A; n = 21) and senior (S; n = 23) horses to touch the target at T10 in three consecutive attempts (a, b, c).

**Figure 3 animals-14-03073-f003:**
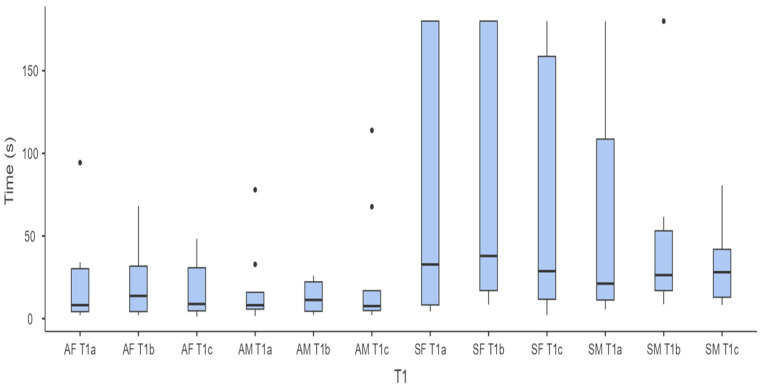
Box-and-whiskers plot showing the time taken by adult mares (AF; n = 12), senior females (SF; n = 15), adult males (AM; n = 9), and senior males (SM; n = 8) to touch the target at T1 across the three consecutive attempts (a, b, c).

**Figure 4 animals-14-03073-f004:**
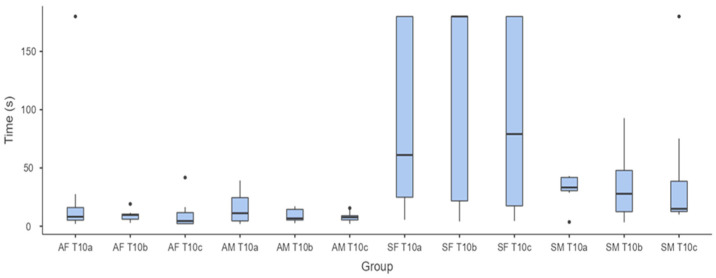
Box-and-whiskers plot showing the time taken by adult females (AF; n = 12), senior females (SF; n = 15), adult males (AM; n = 9), and senior males (SM; n = 8) to touch the target at T10 across the three consecutive attempts (a, b, c).

**Table 1 animals-14-03073-t001:** Signaling data from horses recruited for behavioral testing.

Adult Horses			Senior Horses		
Age	Sex	Breed	Age	Sex	Breed
5	Mare	Pony N.D.	17	Mare	Sella italiano
6	Mare	Hanoverian	17	Mare	Appaloosa
7	Mare	Zangersheide	17	Gelding	N.D.
7	Mare	Sella italiano	17	Mare	Zangersheide
9	Mare	Pony N.D.	18	Gelding	N.D.
9	Mare	Sella italiano	18	Gelding	Gidran
9	Mare	Pure Spanish Horse	18	Gelding	Sella italiano
10	Gelding	N.D.	18	Gelding	Hanoverian
10	Gelding	Pure Spanish Horse	19	Gelding	Arabian
10	Gelding	KWPN	19	Mare	Sella italiano
10	Gelding	N.D.	20	Mare	Maremmano
10	Mare	N.D.	21	Mare	Sella italiano
11	Mare	Murgese	22	Mare	Maremmano
11	Mare	Austr. Warmblood	22	Mare	N.D.
12	Gelding	N.D.	23	Mare	Sella italiano
13	Mare	N.D.	23	Mare	N.D.
13	Gelding	N.D.	24	Mare	N.D.
13	Gelding	Pony N.D.	24	Gelding	N.D.
14	Mare	N.D.	24	Mare	KWPN
14	Gelding	N.D.	25	Mare	Quarter horse
15	Gelding	Arabian	32	Mare	Maremmano
			33	Mare	Sella italiano
			35	Gelding	Trotter

**Table 2 animals-14-03073-t002:** Paired comparisons (Durbin–Conover) between the times (s) required to complete three different attempts (a, b, c) at T10.

Adults	vs.	Seniors	F	*p*
T10a		T10a	4.416	<0.001
T10b		T10b	4.363	<0.001
T10c		T10c	4.994	<0.001

## Data Availability

The original contributions presented in this study are included in the article; further inquiries can be directed to the corresponding author.

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
