# Peer review of "The Memory Abilities of the Elderly Horse"

_animals, 2024, doi:10.3390/ani14213073_

Round 1

Reviewer 1 Report

Comments and Suggestions for Authors

The methods use latency as opposed to the number of correct responses. Given the learning paradigm in this study, latency is as much a measure of motivation as it is of learning/memory. Did any of the animals fail to respond to the visual cue during the T10 testing phase?  This needs to be recorded and the statistical analysis carried out on these data between the two groups. At a maximum of 3 trails however, there is may not be sufficient variation between groups for meaningful statistical analysis. In this context, the study design should have had at least a minimum of 10 trials to create comparative data sets. The paper thus needs to be rewritten in the context of motivation and not learning if latency values are to be used. Future work should focus on using standard cognitive paradigms with an inbuilt significance-based learning criterion e.g. two choice visual discrimination/reversal and number of trails to reach 6 consecutive correct responses.

Figures 1and 2: ages categories need to be presented side by side and in different colour/texture. Legend also doesn’t state the nature of the data- means/medians- nor the nature of the error bars. The number of animals (n) should also be stated.

Table 1. Given the hypotheses being tested, there is no rationale for assessing pairwise comparisons between different time points for different groups of horses

Given the lack of the statistical difference between sexes, there is no real need to present Figures 3 and 4. If they are to be presented, only sex  (and not age) needs to be presented side by side and in different colour/texture. Legend also doesn’t state the nature of the data- means/medians- nor the nature of the error bars. Legend should avoid the term ‘employed’. The number of animals (n) should also be stated.

Line 195 Remove the word ‘horse’

Author Response

Dear Reviewer, 

the require answers are in the attached file. 

Best regards, 

Reviewer 2 Report

Comments and Suggestions for Authors

 Exploring the memory abilities of the elderly horse

The study is designed to study the memory of adult vs older horses to test the hypothesis that as horses age, they as do some other animal species, will show declines in memory. The horses are first clicker trained to reach for a carrot and then are tested to determine whether they show comparative retention after a period of 10 days. As I understand the results, all horses learned the original test and all but two displayed memory for the test after 10 days. The results are discussed to some degree in relation to response speed and memory retention.

Although I commend the authors for addressing an interesting question related to horse memory, I think that the results of this study are under analyzed.

1.     The test is somewhat artificial in that I am uncertain exactly what kind of memory is being tested. There is an associative component to the tests in that that horses are required to learn to associate a clicker sound to a forthcoming presentation of a carrot. In the third phase of the test, the horses are required to respond to cue placed at a distance. This seems to be a spatial test in part. The complexity of the task is not discussed.

2.     I think that the authors expect the reader to understand that horses like carrots. They fail to mention that carrot eating is acquired. Horses show neophobia when first presented with carrots and then become progressively more interested in eating carrots with experience. Have all of the horses had equivalent experience in eating carrots? How was that determined?

3.     There is an attached table of horse age, sex and breed. This should be part of the paper and not an appendix. There are a variety of horse breeds and breed is ignored in the analysis and discussion. There is good documentation on the variability of lifespan associated with horse breeds and this is not discussed in the paper. Although the number of subjects is relatively small for describing age and breed and memory, it is ignored in an analysis of only age. Scatter diagrams and other descriptive and possibly statistical methods could be used to present this information.

4.     As the authors know, as horses age, they develop motor problems, and this is especially true for horses that have engaged performance events. There is no discussion of health in relation to the memory tests. Because the horses have to move to cue in the third phase of the test, motoric health is relevant.

5.     Horses are sensitive to the actions of people, especially actions in related to food presentation. There should be some discussion of the relation between the horse’s interest in the actions of the experimenter, who may or may not be carrying carrots as each trial is given, and the test requirement per se. In addition, the clicker training procedure is interesting, but it is not mentioned whether any of the horses have been subject to any clicker training.

6.     There are many other memory tests that have been given to horses that are less artificial and it would be relevant to discuss whether species-specific behavioral tests might me more sensitive to memory assessment with age.

7.     Even in species that are known to display age related declines in movement and memory (humans and dogs for example), there are vast individual differences. For this reason, the group analysis used in this paper likely represents an under analysis of the potential data. 

Comments on the Quality of English Language

There are some typos and such.

Author Response

Dear Reviewer, 

the required answers are in the attached file.

Best regards, 

Reviewer 3 Report

Comments and Suggestions for Authors

Concluding sentences to abstract = is the slower recovery in fact reduced motivation to engage with the task rather than slower memory recall to engage in a trained task?

Line 50-52 unsubstantiated - where are the data to support this? Needed to support claim in lines 74-76.

Line 57 - is aging also associated with the turning on/off of genes?

Line 58 - what is meant by 'purely biological aspects'? This is not clear from previous information. 

Line 73 - them = horse owners?

Sleep also plays a crucial role in memory formation - why is this not considered in the intro or within methods?

It is surprising to see the lack of review of existing research on equine learning (specifically that pertaining to age) in the introduction. That fact that the literature is now dated either means it is seminal work or has limitations that further support the rationale for the study under review e.g. Equine learning behaviour. Special issue in: Behavioural processes 200776(1); Wolff, A. and Hausberger, M., 1996. Learning and memorisation of two different tasks in horses: the effects of age, sex and sire. Applied animal behaviour science46(3-4), pp.137-143; McLean, A.N., 2004. Short-term spatial memory in the domestic horse. Applied animal behaviour science85(1-2), pp.93-105. But could also benefit from referring to what is currently known about horse memory and learning ability e.g. https://www.sciencedirect.com/science/article/pii/S0168159124001874#sec0010

Line 114-119 = remove reference to tests and abbreviations here they confuse what is written later. Increase repeatability by stating what the time of day and husbandry/ foraging/ exercise/ regimes were. Increase repeatability by stating the equipment used as part of the tests. See Evans et al (2024) as a good example of how to phrase this information. 

Line 121 = assuming this was the box the horse was usually stabled in?

Line 123 = why carrot? Were horses accustomed to receiving a concentrate/hard feed? The sugar content of carrots is much lower than the latter such that horses may not be motivated to engage with the task due to the taste or salience of the reward. Also had hroses been grazing or fed before the study that may also have affected their motivation to engage in the study.

Line 129 = second phase might benefit from a figure or plate?

It is not clear whether ethical approval was acquired to enable this study to go ahead. It is usual to make reference to participant consent or site permission, and to discuss the use of the reward (carrot) with in the horses usual dietary intake. 

Line 184 = which test, do you mean T10 as it looks like some horses took 180 seconds. It might be useful to add the mean duration for each test under adult/senior to table one.

Figure 3 also suggests that female aged horses lacked motivation to engage/ failed to complete the task which may not be memory orientated? Consider works by Briefer Freymond e.g. https://doi.org/10.1016/j.applanim.2014.06.006  and Keison e.g. https://doi.org/10.1016/j.applanim.2020.105075  however there were no significant differences and you conclude to this effect?

The discussion fails to each of the major findings/ limitations in a clear framework. Limited further by the lack of background information provided in the introduction - realistically no new references should appear in the discussion. 

Author Response

(The authors gave the same response as above.)

Round 2

Reviewer 2 Report

Comments and Suggestions for Authors

The authors addressed all of my previous comments except the last comment. 

The could address this comment by including a or some scatter plots in which the score of each animal is displayed. There analysis by sex and breed is ok, but still the data could be mined somewhat more.

Comments on the Quality of English Language

The authors addressed all of my previous comments except the last comment. 

The could address this comment by including a or some scatter plots in which the score of each animal is displayed. There analysis by sex and breed is ok, but still the data could be mined somewhat more.

Author Response

Kind reviewer, the answers are attached.

Reviewer 3 Report

Comments and Suggestions for Authors

I fully commend the amendments made to this paper. However there is still a lack of detail explaining findings in the discussion, please see suggestions below:

Line 74, clarify was it the respondents horses, or was it that respondents of the questionnaire thought that in general horses were pets?

Line 93, suggest adding respectively at the end.

Line 100-101, "The study of memory is therefore very difficult and further complicated..."

Methods = please confirm none of the horses displayed signs of stereotypic/ abnormal repetitive behaviour

Line 144, suggest change to "that the owners occasionally gave to the animals as a reward."

Line 239- 253, feels like an extension of the introduction rather than an explanation of findings?

Discussion - it would be useful to set about explaining why you got each of the results that you did. Currently an explanation of findings is limited to roughly between lines 282 and 295. I appreciate the addition of limitations - it would be good to see the influence of a human observer acknowledged, especially in comparison to an automated system. There is no significant discussion about the strengths and weaknesses of a target touch test as a test of memory in the discussion, and seems quite bias towards the use of the test, even though establishing a test for memory was an aim for the study. As a result the conclusion lacks robustness.

Author Response

(The authors gave the same response as above.)
